# Anti-Neuroinflammation Effect of Standardized Ethanol Extract of Leaves of *Perilla frutescens* var. *acuta* on Aβ-Induced Alzheimer’s Disease-like Mouse Model

**DOI:** 10.3390/pharmaceutics17081045

**Published:** 2025-08-12

**Authors:** Hyunji Kwon, Jihye Lee, Eunhong Lee, Somin Moon, Eunbi Cho, Jieun Jeon, A Young Park, Joon-Ho Hwang, Gun Hee Cho, Haram Kong, Mi-Houn Park, Sung-Kyu Kim, Dong Hyun Kim, Ji Wook Jung

**Affiliations:** 1Department of Advanced Translational Medicine, School of Medicine, Konkuk University, 120 Neungdong-ro, Gwangjin-gu, Seoul 05029, Republic of Korea; hyunje2701@naver.com (H.K.); ans5346@naver.com (S.M.); bee2634@naver.com (E.C.); ji6785@naver.com (J.J.); khg99665@naver.com (A.Y.P.); 2Industry Academic Cooperation Foundation, Daegu Haany University, 1 Hanuidae-ro, Gyeongsan-si 38610, Gyeongsangbuk-do, Republic of Korea; jhlee86@dhu.ac.kr; 3Department of Chemical Engineering, Yeungnam University, 280 Daehak-ro, Gyeongsan-si 38541, Gyeongsangbuk-do, Republic of Korea; lablabuuu@naver.com; 4Borambio Co., Ltd., 119 Dandae-ro, Dongnam-gu, Cheonan 31116, Chungcheongnam, Republic of Korea; jh_hwang@borambio.com (J.-H.H.); joo2413@borambio.com (G.H.C.); hrkong@borambio.com (H.K.); pmh@borambio.com (M.-H.P.); skkim@borambio.com (S.-K.K.); 5Department of Pharmacology, School of Medicine, Konkuk University, 268 Chungwon-daero, Chungju-si 27478, Chungcheongbuk-do, Republic of Korea; 6Research Institute of Medical Science, Konkuk University School of Medicine, 268 Chungwon-daero, Chungju-si 27478, Chungcheongbuk-do, Republic of Korea; 7Department of Cosmetic, Daegu Haany University, 1 Hanuidae-ro, Gyeongsan-si 38610, Gyeongsangbuk-do, Republic of Korea

**Keywords:** cognitive impairment, learning and memory, neuroinflammation, BDNF, *Perilla frutescens* var. *acuta*

## Abstract

**Background/Objectives:** *Perilla frutescens* var. *acuta* Kudo, a member of the Lamiaceae family, has been previously reported to reduce neuroinflammation and potentially decrease Aβ plaque accumulation in 5XFAD mice. In this study, we aimed to evaluate the anti-neuroinflammatory potential of a standardized 60% ethanol extract of Perilla leaves (PE), optimized for commercial application. **Methods**: The inflammatory response was assessed in LPS-stimulated BV2 microglial cells, and the cognitive improvement was evaluated in an AD animal model induced by intracerebroventricular injection of Aβ. **Results**: Using LPS-stimulated BV2 microglial cells and an Aβ-injected ICR mouse model of Alzheimer’s disease, we found that PE significantly suppressed the LPS-induced production of nitric oxide and pro-inflammatory mediators, including IL-6, TNF-α, NF-κB, iNOS, and COX-2, along with inhibition of JNK and p38 MAPK activation. Furthermore, PE upregulated CREB and BDNF expression. In vivo, PE administration alleviated Aβ-induced cognitive deficits, which were associated with reduced expression of JNK, NF-κB, iNOS, and COX and increased CREB/BDNF signaling in the hippocampus. Behavioral assessments—including passive avoidance, Morris water maze, novel object recognition, and Y-maze tests—confirmed the improvement in cognitive function. **Conclusions**: Collectively, these findings demonstrate that PE exerts significant anti-neuroinflammatory and neuroprotective effects, supporting its potential as a functional ingredient for cognitive enhancement.

## 1. Introduction

Alzheimer’s disease (AD) is the most prevalent neurodegenerative and dementia-causing disease of our time, characterized by progressive cognitive decline, memory loss, and behavioral changes. Pathologically, AD features amyloid-β (Aβ) plaques, neurofibrillary tangles of tau protein, and neuronal degeneration, but increasing evidence points to a pivotal role of immune processes in its pathogenesis [1]. Despite extensive research efforts, AD remains largely irreversible once clinical symptoms manifest, emphasizing the critical need for early intervention strategies [2]. Current treatments primarily focus on managing symptoms rather than preventing or curing the condition, highlighting the urgent need for novel therapeutic approaches [3].

Mild cognitive impairment (MCI) represents a transitional state between normal cognitive aging and dementia, with approximately 10–15% of individuals aged 65 and above being affected [4]. MCI, particularly the amnestic subtype (aMCI), is often considered a prodromal stage of AD, characterized by subtle cognitive deficits that do not significantly interfere with daily activities [5,6]. Unlike established AD, MCI presents a crucial window of opportunity for therapeutic intervention, as early detection and treatment may prevent or delay progression to dementia [7]. This makes MCI an important target for preventive strategies and early therapeutic interventions [8].

Neuroinflammation plays a crucial role in the pathogenesis of both Alzheimer’s disease (AD) and mild cognitive impairment (MCI) [9], as supported by experimental, epidemiological, and genetic evidence [10,11]. It is marked by activation of microglia and astrocytes, leading to the release of pro-inflammatory cytokines and reactive oxygen species [12]. Notably, neuroinflammation appears early in disease progression and co-localizes with tau pathology in MCI, suggesting it is a driver rather than a consequence of neurodegeneration [13,14]. Various cell types, including microglia, astrocytes, oligodendrocytes, lymphocytes, and peripheral myeloid cells, contribute to this complex inflammatory environment [15,16]. These insights have spurred growing interest in anti-inflammatory therapies for AD and MCI.

*Perilla frutescens* var. *acuta* Kudo, commonly known as “Nga-mon” in Thailand or “Soyeop” in Korea, is a member of the mint family (Lamiaceae) traditionally used across Asia for its culinary and medicinal properties, including treatment of respiratory and neurological conditions [17,18]. Its pharmacological effects are attributed to bioactive compounds such as polyphenols, flavonoids, and essential oils [2], with rosmarinic acid identified as a major active constituent [9]. The 60% ethanol extract of *Perilla* contains particularly high levels of rosmarinic acid and scutellarin, both known for antioxidant, anti-inflammatory, and neuroprotective activities [19]. Rosmarinic acid, in particular, modulates microglial activation and suppresses inflammatory mediators like TNF-α and IL-1β, while also inhibiting apoptosis and enhancing neuronal survival [20,21]. Recent studies have suggested that *Perilla* extracts exert their effects through pathways including c-Jun, NF-κB, and Akt [19], which are closely linked to neuroinflammatory mechanisms underlying MCI and AD [5,22]. These findings support the potential of *Perilla* as a therapeutic candidate for cognitive impairment associated with neuroinflammation and oxidative stress [15].

Therefore, the present study aims to investigate the anti-neuroinflammatory and neuroprotective effects of 60% ethanol extract from *Perilla frutescens* var. *acuta* kudo and its potential to improve cognitive function. By elucidating the mechanisms through which this extract modulates neuroinflammatory pathways and protects neuronal integrity, this research may provide valuable insights into novel therapeutic approaches for MCI and potentially delay progression to AD.

## 2. Materials and Methods

### 2.1. Materials

Donepezil was purchased from Sigma-Aldrich (St. Louis, MO, USA). The *Perilla frutescens* extract (PE) used in this study was generously supplied by Boram Bio Co., Ltd. (Cheonan, Republic of Korea) and manufactured at the S&D production facility located in Chungcheongbuk-do, Republic of Korea. Two independent batches of PE were utilized, identified as Lot No. 210915-001 and Lot No. 240116-001, corresponding to their respective production dates. The extract was prepared by subjecting *Perilla frutescens* leaves to extraction with 60% ethanol at 50 °C for 24 h, followed by spray-drying. The rosmarinic acid content of the resulting extract was quantified as 12.7 mg/g (Appendix A). The mouse IL-6 and TNF ELISA kit and mouse monoclonal iNOS antibody were purchased from BD biosciences (Franklin Lakes, NJ, USA). Mouse monoclonal COX-2, rabbit polyclonal phosphorylated p38 (pp38), mouse monoclonal phosphorylated NF-κB p65, rabbit polyclonal NF-κB p65, mouse monoclonal β-actin, mouse monoclonal Bcl-2, and rabbit polyclonal Bax antibodies were obtained from Santa Cruz Biotechnology (Dallas, TX, USA). Rabbit polyclonal phosphorylated SAPK/JNK (pJNK), rabbit polyclonal SAPK/JNK, rabbit polyclonal p38 MAPK, rabbit polyclonal cleaved caspase-3, rabbit polyclonal caspase-3, rabbit monoclonal phosphorylated CREB (pCREB), rabbit monoclonal CREB antibodies, and RIPA buffer were purchased from Cell Signaling Technology (Danvers, MA, USA). Rabbit monoclonal BDNF antibody was obtained from Cusabio (Houston, TX, USA), and rabbit polyclonal antibodies against histone H3 and PARP1 were purchased from Epitomics (Abcam, Waltham, MA, USA). Mouse monoclonal GAPDH antibody was obtained from NKMAXBio (Seongnam, Republic of Korea). Goat anti-mouse IgG polyclonal antibody was purchased from GeneTex (Irvine, CA, USA), and goat anti-rabbit IgG polyclonal antibody was obtained from Enzo Life Sciences (Farmingdale, NY, USA). The Pierce^TM^ BCA Protein Assay Kit, NE-PER^TM^ Nuclear and Cytoplasmic Extraction Reagents, and SuperSignal^TM^ West Femto Maximum Sensitivity Substrate were purchased from Thermo Fisher Scientific (Waltham, MA, USA). WestGlowTM PICO PLUS Chemiluminescent Substrate was obtained from BIOMAX (Seoul, Republic of Korea). Dulbecco’s Modified Eagle Medium (DMEM) was obtained from Welgene (Gyeongsan, Republic of Korea), and fetal bovine serum (FBS) was purchased from Gibco (Waltham, MA, USA). The CellTiter 96^®^ AQ_ueous_ One Solution Cell Proliferation Assay (MTS) was purchased from Promega (WI, USA). All other materials were obtained from standard commercial suppliers and were of the best quality available.

### 2.2. Cell Culture and LPS-Induced BV2 Activation

The murine microglial cell line BV2 was cultured in DMEM supplemented with 10% FBS and 1% penicillin–streptomycin under conditions of 37 °C and 5% CO_2_ with humidity.

To evaluate the neuroinflammatory effects of the PE extract, BV2 cells were stimulated with LPS (50 ng/mL) for 24 h to activate microglia.

### 2.3. Cell Viability Assay

Microglial cell viability was assessed using the MTS assay. BV2 cells were seeded at a density of 1 × 10^5^ cells/mL in 100 μL per well in a 96-well plate and incubated for 24 h. After removing the culture medium, serum-free medium was added, followed by treatment with various concentrations of the test substance or pretreatment with 50–1000 μg/mL for 2 h followed by LPS treatment at 50 ng/mL for 24 h. Then, 10 μL of MTS solution was added to each well and incubated for 1 h. Absorbance was measured at 490 nm using a microplate reader (Tecan, Männedorf, Switzerland). Cell viability was expressed as the percentage decrease in absorbance compared to untreated control wells.

### 2.4. Measuring NO Production

To measure the nitrite level, Griess reagent was used. BV2 cells were seeded in a 24-well plate and pretreated with PE extract (50, 100, 300, and 500 μg/mL) for 2 h, followed by stimulation with LPS for 24 h. After incubation, an equal volume of cell supernatant and Griess reagent was mixed, and the absorbance was measured at 540 nm using a microplate reader.

### 2.5. Western Blot

The cells were collected and lysed using RIPA buffer supplemented with protease and phosphatase inhibitors to extract total protein for analysis of general protein expression. For detection on nuclear NF-κB, nuclear proteins were separately extracted using a nuclear and cytoplasmic extraction kit, according to the manufacturer’s instructions. After centrifugation, the cell lysate was collected. Protein concentrations were determined using a BCA protein assay kit, and the samples were separated by SDS-PAGE, transferred to a PVDF membrane, and blocked with 5% skim milk. Membranes were incubated with primary antibodies overnight at 4 °C, followed by secondary antibodies. Protein bands were visualized using an ECL detection kit and quantified using ImageJ software (Ver. 1.53, NIH, Bethesda, MD, USA; accessed on 30 December 2024). β-actin and GAPDH were used as the loading controls for total protein, and histone H3 was used as the nuclear protein loading control.

### 2.6. ELISA—Measuring Cytokine Release

To measure the concentrations of TNF-α and IL-6 in the supernatant of treated BV2 cells, an ELISA kit was used. According to the manufacturer’s instructions, a capture antibody was added to a 96-well plate and coated overnight at 4 °C. Afterward, each well was washed and blocked with 10% FBS at room temperature for 1 h. Diluted standards and supernatants were incubated for 2 h. After incubation with the detection antibody for 1 h, the wells were treated with SAv-HRP enzyme reagent and then reacted with TMB substrate solution for color development. The reaction was stopped with 1M H_3_PO_4_, and the absorbance was measured at 450 nm using a microplate reader.

### 2.7. Animals

Experimental animals (male ICR mice, 4 weeks old) were purchased from DBL (Eumseong, Republic of Korea). The mice were acclimated to the animal facility for one week before the experiments began. The animal facility maintained a 12-h light–dark cycle (7:00 AM/7:00 PM), and temperature (23 ± 1 °C) and humidity (60 ± 5%) were kept constant. During both the acclimation and experimental periods, the animals had ad libitum access to food and water. All animal experiments were conducted in accordance with NIH guidelines [23] and were approved by the Animal Ethics Committee of Daegu Haany University (DHU2021-095, DHU2025-017).

Aβ_1-42_ was dissolved in PBS at a concentration of 1 mg/mL (222 μM) and incubated at 37 °C with gentle shaking for 24 h to allow aggregation. Then, 5 μL of the solution was injected into the intracerebroventricular (ICV) region. For the injection, mice were anesthetized with isoflurane (3% for induction and 1.5% for maintenance), and Aβ_1-42_ aggregates were injected into the third ventricle using stereotaxic coordinates (AP: −2.00 mm, ML: 0 mm, DV: −2.00 mm). From the following day, PE was administered orally once daily at doses of 100, 250, or 500 mg/kg, which continued until the end of the experiment. The doses (100, 250, and 500 mg/kg) were selected based on previous studies [24]. Each experimental group consisted of 9–10 mice. On day 8, behavioral tests were sequentially conducted in the following order: Y-maze test, novel object recognition test, passive avoidance test, and Morris water maze test. For Western blot analysis, animals were sacrificed on day 7, and hippocampal tissue was collected. To eliminate potential bias, a blinded experimental design was employed, in which the investigators were unaware of the group allocations during data collection and analysis.

### 2.8. Y-Maze Test

A Y-maze instrument consisting of three arms was used (20 cm long × 10 cm wide × 20 cm high) at 120° angles. The experimental animal was placed in one arm, and entry into the three arms was recorded for 8 min. A score was given when the experimental animal entered a new branch instead of the previous branch. Spontaneous alternation was calculated by the following formula. Spontaneous alternation = (score/(total arm entry − 2)) × 100.

### 2.9. Novel Object Recognition Test

The novel object recognition test was performed in an open field box measuring 40 × 40 × 30 cm. Prior to testing, two identical objects were placed 5 cm away from the walls of the box. Following drug administration, the mouse was positioned at the center of the box and allowed to explore the objects for 5 min. After a 24-h retention interval, one of the original objects was replaced with a novel object, and the mouse was reintroduced to the center of the box for another 5-min exploration session. During each session, the time spent engaging in exploratory behaviors—such as sniffing, touching, or licking—toward the familiar and novel objects was recorded. The percentage of exploration time for each object was calculated, and a higher preference for the novel object was interpreted as an indicator of improved learning and memory function.

### 2.10. Passive Avoidance Test

The passive avoidance box is divided into a dark room and a light room. The experimental animal is placed in the light room, and an electric shock is applied as it crosses into the dark room. The next day, when the animal is placed back in the light room, it remembers the electric shock from the dark room and stays in the light room. The time spent in the light room is measured to assess memory.

### 2.11. Morris Water Maze Test

To evaluate hippocampus-dependent spatial learning and long-term memory, the Morris water maze test was conducted using a circular pool with a diameter of 90 cm and a height of 45 cm. A platform (9 cm in diameter, 25 cm in height) was placed in one of the quadrants, and the pool was filled with clean water (20 ± 2 °C) to a level approximately 1 cm above the platform surface. Four visual cues were placed around the pool to divide it into quadrants, and the entry point was varied in each trial. Mice were trained four times per day for four consecutive days, with each trial lasting up to 60 s. If a mouse located the hidden platform within 60 s, the trial was concluded. If not, the mouse was guided to the platform and allowed to remain there for 10 s.

On the final day, the platform was removed to assess memory retention, and the time spent in the target quadrant (where the platform had previously been located) was measured for 60 s. All trials were recorded and analyzed using the EthoVision software (Ver 3.1.16, Noldus, Netherlands; accessed on 27 October 2021).

### 2.12. Statistics

Data analysis was conducted using GraphPad Prism (Ver. 8, San Diego, CA, USA; accessed on 26 February 2025) statistical software. For comparisons involving three or more groups, one-way ANOVA was used, followed by post hoc tests using the Newman–Keuls test. For analysis of the training sessions in the Morris water maze test, two-way ANOVA (factors: day and group) was performed, followed by Bonferroni post hoc tests. Animals were excluded if they exhibited minimal activity during the tests, defined as follows: total arm entries ≤ 5 in the Y-maze test, object exploration time ≤ 5 s in the object recognition test, step-through latency ≥ 60 s in the passive avoidance test, and floating without active swimming in the Morris water maze. All data are presented as means ± SDs. Statistical significance was indicated when *p* < 0.05.

## 3. Results

### 3.1. The Effect of PE on LPS-Induced Release of Inflammatory Factors

To investigate the anti-neuroinflammatory effects of PE, we treated BV2 microglial cells with LPS and measured the secretion levels of inflammation-related factors, including NO, IL-6, and TNF-α. LPS treatment significantly increased NO release (F_5, 66_ = 685.8, *p* < 0.001, n = 12/group, Figure 1A) as well as the secretion of IL-6 (*F*_5, 18_ = 75.96, *p* < 0.001, n = 4/group, Figure 1B) and TNF-α (*F*_5, 18_ = 31.89, *p* < 0.001, n = 4/group, Figure 1C) in BV2 cells, indicating the induction of neuroinflammation. However, in the group pretreated with PE prior to LPS stimulation, NO (*p* < 0.05, Figure 1A), IL-6 (*p* < 0.05, Figure 1B), and TNF-α (*p* < 0.05, Figure 1C) secretion was significantly reduced in a concentration-dependent manner compared to the only-LPS-treated group. These results suggest that PE effectively inhibits LPS-induced neuroinflammation.

### 3.2. The Effect of PE on LPS-Induced Signaling Changes

The release of NO and various inflammation-related cytokines induced by LPS is attributed to the activation of iNOS and COX-2. Therefore, we examined the effect of PE on the LPS-induced upregulation of iNOS and COX-2 expression. In BV2 cells treated with LPS, iNOS (*F*_5, 28_ = 49.47, *p* < 0.001, n = 5–6/group, Figure 2A) and COX-2 (F_5, 23_ = 18.42, *p* < 0.001, n = 4–5/group, Figure 2B) levels were significantly increased. However, in the group pretreated with PE prior to LPS stimulation, iNOS (*p* < 0.05, Figure 2A) and COX-2 (*p* < 0.01, Figure 2B) levels were significantly reduced in a concentration-dependent manner compared to the LPS-only treated group. These findings suggest that PE may regulate neuroinflammation by modulating iNOS and COX-2 expression.

The expression of iNOS and COX-2 induced by LPS is mediated by the MAPK signaling pathway and the nuclear translocation of NF-κB downstream of TLR4 signaling [25]. Therefore, we investigated the effect of PE on the activation of JNK and p38 and nuclear translocation of NF-κB in response to LPS stimulation. In BV2 cells treated with LPS, pJNK (*F*_5, 24_ = 15.07, *p* < 0.001, n = 5/group, Figure 2C) and pp38 (*F*_5, 28_ = 15.70, *p* < 0.001, n = 5–6/group, Figure 2D) levels were significantly increased. Moreover, nuclear level of NF-κB (*F*_5, 28_ = 11.27, *p* < 0.001, n = 5–6/group, Figure 2E) was also significantly increased in the LPS-treated group. However, in the group pretreated with PE prior to LPS stimulation, pJNK (*p* < 0.05, Figure 2C) and pp38 (*p* < 0.05, Figure 2D) levels and NF-κB nuclear levels (*p* < 0.05, Figure 2E) were significantly reduced compared to the only-LPS-treated group. These findings suggest that PE may regulate neuroinflammation by modulating MAPK signaling.

### 3.3. The Effect of PE on LPS-Induced CREB and BDNF Expression

LPS treatment is known to affect microglial BDNF production, which is known to play crucial roles in synaptic plasticity, learning and memory, and neuroprotection [26]. Therefore, we investigated the effect of PE on CREB and BDNF expression, which are suppressed by LPS. In BV2 cells treated with LPS, the pCREB (*F*_5, 21_ = 13.84, *p* < 0.001, n = 4–5/group, Figure 3A) level was significantly decreased. However, in the group pretreated with PE (300, 500 μg/mL) prior to LPS stimulation, pCREB (*p* < 0.01, Figure 3A) levels were significantly increased compared to the only-LPS-treated group. In BV2 cells treated with LPS, mature BDNF was significantly decreased (*F*_5, 15_ = 7.15, *p* < 0.05, n = 3–5/group, Figure 3B). However, pretreatment with PE (300 and 500 μg/mL) significantly increased mature BDNF levels. These findings indicate that PE may improve LPS-induced reductions in CREB/BDNF signaling.

### 3.4. Effect of PE on Aβ-Induced Cognitive Impairments

To confirm if PE improves Aβ-induced cognitive impairments, we conducted the Y-maze test, novel object recognition test, passive avoidance test, and Morris water maze test.

#### 3.4.1. Y-Maze Test

In the Y-maze test used to assess short-term memory, the Aβ-injected control group showed a significant reduction in spontaneous alternation compared to the normal group (*F*_5, 48_ = 5.72, *p* < 0.001, n = 9/group; Figure 4A). Treatment with PE significantly increased spontaneous alternation, and the Donepezil group also showed significant improvement (*p* < 0.01). The total number of arm entries did not significantly differ among groups, indicating that the treatments did not affect locomotor activity (Figure 4B).

#### 3.4.2. Novel Object Recognition Test

In the novel object recognition test, the Aβ-injected control group showed a significantly lower preference ratio for the novel object compared to the normal group, indicating impaired recognition memory (*F*_5, 48_ = 8.827, *p* < 0.001, n = 9/group; Figure 5A). Post-treatment with PE (100, 250, or 500 mg/kg) significantly increased the preference ratio (*p* < 0.01), suggesting improvement in object recognition behavior despite prior Aβ-induced cognitive impairment. Likewise, the discrimination index was also significantly decreased in the Aβ-injected control group compared to the normal group (*F*_5, 48_ = 11.75, *p* < 0.001, n = 9/group; Figure 5B). In contrast, mice administered PE (100, 250, or 500 mg/kg) showed a significantly increased discrimination index compared to the control group (*p* < 0.01).

#### 3.4.3. Passive Avoidance Test

The Aβ injection significantly decreased step-through latency in the retention trial (*F*_5, 54_ = 32.29, *p* < 0.001, n = 10/group, Figure 6), indicating that Aβ induced a decline in learning and memory. In the acquisition trial, no significant difference in step-through latency was observed among any of the experimental groups (*F*_5, 54_ = 0.3913, *p* > 0.05, n = 10/groups). However, in the retention trial, PE-treated groups (100, 250, or 500 mg/kg, p.o.) showed no cognitive impairment induced by Aβ (*p* < 0.001), indicating that pretreatment with PE effectively prevented Aβ-induced memory deficits. Donepezil, a positive drug, also blocked Aβ -induced cognitive impairment in the passive avoidance test.

#### 3.4.4. Morris Water Maze

In the Morris water maze learning test (Figure 6A), there was no significant difference in escape latency among groups on day 1. However, by day 4, the Aβ-injected control group exhibited significantly prolonged latency (40.00 ± 10.5 s) compared to the normal group (24.81 ± 4.4 s, *p* < 0.01, Figure 7A), whereas the PE-treated groups showed significantly shorter latencies compared to the Aβ-injected control group (100 mg/kg, 26.17 ± 14.15 s; 250 mg/kg, 24.9 ± 12.5 s; 500 mg/kg, 22.72 ± 6.5 s, *p* < 0.01, Figure 7A). The 500 mg/kg PE group demonstrated an effect comparable to the Donepezil group (19.53 ± 5.1 s, Figure 7A). Two-way ANOVA revealed that there was a significant group effect (group, *F*_5, 192_ = 12.89, *p* < 0.001, n = 9/group; day, *F*_3, 192_ = 58.14, *p* < 0.001, n = 9/group; interaction, *F*_15, 192_ = 0.6340, *p* > 0.05, n = 9/group; Figure 7A).

In the probe trial on day 5 (Figure 7B), the time spent in the target quadrant was significantly reduced in the Aβ-injected control group (15.49 ± 1.6 s) compared to the normal group (24.83 ± 5.7 s, *p* < 0.001), while PE-treated groups spent significantly more time compared to the Aβ-injected control group (100 mg/kg, 18.61 ± 4.2 s; 250 mg/kg, 22.62 ± 3.4 s; 500 mg/kg, 24.28 ± 4.3 s, *F*_5, 48_ = 9.263, *p* < 0.001, n = 9/group; Figure 7B). The Donepezil group also showed recovery (24.71 ± 2.0 s, Figure 7B), indicating that PE improves learning ability and spatial memory in the Aβ-induced cognitive impairment model.

### 3.5. Effect of PE on Aβ-Induced Neuroinflammation

The injection of Aβ activates neuroinflammation, which is associated with cognitive impairment. Therefore, we examined whether the anti-neuroinflammatory effects of PE observed in in vitro experiments could be replicated in an in vivo model. As in previous studies, ICV injection of Aβ significantly increased iNOS (*F*_2, 6_ = 13.56, *p* < 0.01, n = 3/group, Figure 8A) and COX-2 (*F*_2, 6_ = 10.66, *p* < 0.05, n = 3/group, Figure 8B) expression. PE treatment significantly reduced iNOS (*p* < 0.01, Figure 8A) and COX-2 (*p* < 0.05, Figure 8B) expression.

Regarding intracellular signaling, Aβ administration significantly activated JNK (*F*_2, 6_ = 85.18, *p* < 0.001, n = 3/group, Figure 8C) and p38 (*F*_2, 6_ = 8.574, *p* < 0.05, n = 3/group, Figure 8D) and increased pNF-κB levels (*F*_2, 6_ = 33.06, *p* < 0.001, n = 3/group, Figure 8E). However, PE treatment led to a marked reduction in JNK (*p* < 0.001, Figure 8C) activation and pNF-κB (*p* < 0.001, Figure 8E) levels compared to the Aβ-treated control group. These findings suggest that PE may suppress certain neuroinflammation-related signaling pathways in an in vivo model.

### 3.6. Effect of PE on Aβ-Induced CREB and BDNF Expression

In the hippocampus of Aβ-injected mice, an evident reduction in the levels of pCREB (*F*_2, 6_ = 5.385, *p* < 0.05, n = 3/group; Figure 9A) was observed. And the PE-treated group showed a significant increase in the level of pCREB compared to the Aβ-injected control group (*p* < 0.04, Figure 9A). However, BDNF levels showed a decreasing trend in the Aβ-treated group and an increasing trend in the PE-treated group; however, these changes were not statistically significant (*F*_2, 6_ = 2.975, *p* > 0.05, n = 3/group; Figure 9B). These results suggest that PE may regulate Aβ-induced downregulation of these neuroplasticity-related proteins.

## 4. Discussion

In this study, we demonstrated that PE exerts significant anti-neuroinflammatory and neuroprotective effects, which in turn translate into improved cognitive function in both cellular and animal models of AD. In LPS-stimulated BV2 microglial cells, PE significantly suppressed the excessive production of inflammatory mediators. In particular, it inhibited the expression of iNOS, leading to a concentration-dependent reduction in NO levels. Additionally, PE downregulated the expression of COX-2 and the pro-inflammatory cytokines NF-κB, IL-6, and TNF-α. It also inhibited the activation of JNK and p38 MAPK—key components of the MAPK pathway, which acts as an upstream regulator of inflammatory gene expression.

These findings indicate that PE reduces LPS-induced inflammatory responses. Mechanistically, the extract’s anti-inflammatory action was associated with the inhibition of stress-activated MAPKs, particularly JNK and p38 MAPK, which are key upstream regulators of inflammatory gene expression. Such an effect is in line with previous reports where PE suppressed LPS-induced NO, IL-6, and TNF-α production in microglia by downregulating the MAPK/iNOS pathway [27]. Given that microglial-mediated neuroinflammation is a prominent driver of AD pathogenesis and NF-κB, IL-6, and TNF-α levels are found to be elevated in AD patient brains [28], the ability of PE to regulate the expression of these cytokines and enzymes suggests its potential to alleviate AD-related neuroinflammatory damage and mitigate mild cognitive impairment, a prodromal stage of Alzheimer’s disease. Notably, our results concur with Kang et al. (2022), who showed that *P. frutescens* leaf extract attenuated microglial activation and its downstream inflammatory signaling in a neurodegeneration model [27]. By blunting excessive NO and cytokine release, PE may reduce the neurotoxic effects of reactive nitrogen species and cytokine cascades that otherwise contribute to synaptic dysfunction and neuron loss in AD.

Furthermore, PE boosted levels of key synaptic plasticity-related proteins, namely CREB and BDNF. CREB is a transcription factor essential for memory formation and neuronal survival, in part by driving expression of BDNF [29]. In our study, PE-treated mice showed enhanced CREB/BDNF signaling in the hippocampus, indicating activation of pro-survival signaling. This aligns with the current understanding that Aβ pathology can downregulate CREB/BDNF signaling, contributing to synaptic loss and cognitive impairment [30]. By restoring the CREB/BDNF pathway, PE may promote synaptic resilience. Our results echo findings with other plant-derived treatments; for instance, Du et al. (2020) [31] reported that an herbal polysaccharide ameliorated Aβ-induced memory deficits by activating the BDNF/TrkB/CREB cascade, which concurrently inhibited neuronal apoptosis. Thus, PE’s dual action in curbing inflammatory damage and bolstering neurotrophic support provides a strong mechanistic basis for neuroprotection in the AD context.

In BV2 microglial cells, LPS stimulation induced a strong inflammatory response, characterized by the upregulation of pro-inflammatory cytokines and activation of signaling pathways such as NF-κB and MAPKs. This inflammatory milieu is known to promote oxidative stress and mitochondrial dysfunction, ultimately triggering apoptosis. In our study, PE significantly suppressed the LPS-induced production of inflammatory mediators and inhibited the activation of JNK and p38 MAPK pathways. Alongside these anti-inflammatory effects, PE also attenuated LPS-induced cell death, as evidenced by reduced levels of cleaved caspase-3 and PARP-1, and the restoration of Bcl-2 expression (Appendix A). These results suggest that the anti-apoptotic effect of PE is closely linked to its ability to suppress the inflammatory response. By dampening inflammation, PE may mitigate downstream pro-apoptotic signaling, thereby protecting BV2 cells from LPS-induced cytotoxicity.

The neuroprotective effects of PE at the molecular level were reflected in pronounced improvements in cognitive function across multiple behavioral assays. In the Morris water maze test of spatial learning and memory, Aβ-injected mice treated with PE learned to locate the hidden platform faster and spent more time in the target quadrant during probe trials than vehicle-treated Aβ controls, indicating preservation of hippocampal-dependent spatial memory. Similarly, PE significantly improved spontaneous alternation in the Y-maze test, demonstrating an enhancement in working memory and cognitive flexibility. In the passive avoidance paradigm, PE-treated mice showed longer latency to enter the shock-associated compartment on the retention trial, which signifies better retention of aversive memory. PE also enhanced recognition memory in the novel object recognition test, as evidenced by a higher discrimination index (greater exploration of the novel object). These behavioral findings strongly support the cognitive-enhancing potential of PE. They agree with previous studies on Perilla extracts and rosmarinic acid in AD models. Lee et al. (2016) found that oral administration of *P. frutescens* extract (50 mg/kg) or its active component rosmarinic acid significantly reversed Aβ-induced cognitive impairment in multiple tasks, including spontaneous alternation (T-maze), object recognition, and the Morris water maze [32]. Treated mice in that study showed shorter escape latencies and increased platform crossings in the water maze, paralleling our observations. Importantly, the cognitive benefits of PE are tightly linked to the underlying biochemical changes. By reducing neuroinflammation and oxidative stress in the brain (as evidenced by lower NO and malondialdehyde levels [33]) and by preventing neuronal apoptosis, PE likely preserved functional neurons and synaptic connectivity in the hippocampus, which enabled treated animals to perform significantly better in learning and memory tests. Consistent with this idea, the fermented Perilla extract used by Seo et al. (2024) improved Y-maze alternation and passive avoidance performance in a sleep deprivation cognitive impairment model, an effect accompanied by restoration of hippocampal BDNF levels and CREB activation [22]. Together, these findings suggest that the cognitive improvements conferred by PE are a direct consequence of its multimodal mitigation of AD-related neuropathology—dampening inflammation and apoptosis while promoting synaptic plasticity.

Our results are further reinforced by comparisons with the prior literature on Perilla extracts, specifically rosmarinic acid, a major phenolic constituent of Perilla leaves. The 60% ethanol extract used in our study was rich in rosmarinic acid, which is known for its potent antioxidant and anti-inflammatory properties [32]. Rosmarinic acid has been shown to exert neuroprotective effects by suppressing microglial activation, promoting M2 polarization, reducing pro-inflammatory cytokine production, and downregulating inflammatory pathways such as NF-κB and MAPKs, including JNK and p38 [20,34]. A recent study further demonstrated that rosmarinic acid administration in a triple-transgenic AD model alleviated Aβ plaque formation and tau hyperphosphorylation, resulting in cognitive improvement [34]. These mechanisms are consistent with our findings, where PE inhibited JNK activation and upregulated CREB–BDNF signaling. Moreover, rosmarinic acid has been reported to enhance monoaminergic signaling, potentially interfering with Aβ aggregation [35]. Collectively, this evidence provides a strong rationale for the efficacy of our rosmarinic acid-enriched *Perilla* extract in modulating multiple pathological targets in AD. However, a comprehensive LC-MS/MS analysis of Perilla frutescens var. acuta leaves identified 22 phenolic compounds, with high levels of rosmarinic acid (57,019.6–66,608.2 ng/g) as well as significant quantities of luteolin and its derivatives, quercetin glycosides, caffeic acid, and others. These compounds are all documented in the literature for anti-neuroinflammatory and antioxidant effects. Therefore, the observed efficacy of our standardized extract cannot rule out the possibility that it results from synergistic and/or additive pharmacological actions of rosmarinic acid, luteolin, and other co-extracted phytochemicals.

A key strength of this study is the use of a standardized PE extract, ensuring consistent levels of rosmarinic acid and other bioactives, such as luteolin and apigenin glycosides. *Perilla frutescens* has a long history of use as both a traditional medicinal and culinary herb in East Asia [36], and PE was well tolerated in our study with no adverse effects observed.

Despite these promising outcomes, several limitations should be acknowledged. Our AD model involved acute Aβ injection, which does not fully recapitulate the chronic, progressive pathology of human AD. Future research using transgenic models (e.g., APP/PS1 or 3xTg-AD) with longer treatment periods is needed to evaluate PE’s effects on amyloid and tau pathology. In addition, we did not assess pharmacokinetic parameters such as RA bioavailability or brain penetration, which should be addressed in future studies. Finally, although rosmarinic acid appears to be a principal active constituent, PE contains multiple phytochemicals, and their potential synergistic effects warrant further investigation using fractionation or rosmarinic acid-equivalent comparisons.

## 5. Conclusions

In conclusion, this study provides a comprehensive demonstration that Perilla frutescens var. acuta extract can counteract AD-related neuroinflammation and neuronal injury, thereby enhancing cognitive performance and improving mild cognitive impairment (MCI).

## Figures and Tables

**Figure 1 pharmaceutics-17-01045-f001:**
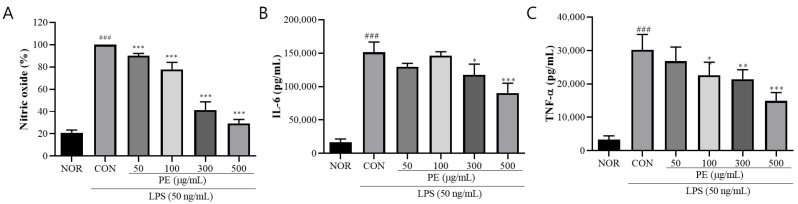
PE on LPS-induced release of neuroinflammation-related factors. (**A**) BV2 microglial cells were pretreated with PE at various concentrations (50, 100, 300, and 500 μg/mL) for 2 h, followed by stimulation with LPS (50 ng/mL) for 24 h. NO production was measured in the culture supernatant using the Griess reagent assay. (**B**) IL-6 levels in the supernatant were quantified by ELISA after PE pretreatment and subsequent LPS stimulation. (**C**) TNF-α levels were also measured using ELISA under the same treatment conditions. Data are presented as means ± SDs. Statistical analysis was performed using one-way ANOVA followed by the Newman–Keuls post hoc test. ### *p* < 0.001 vs. normal (NOR) group; * *p* < 0.05, ** *p* < 0.01, and *** *p* < 0.001 vs. LPS-treated control (CON) group.

**Figure 2 pharmaceutics-17-01045-f002:**
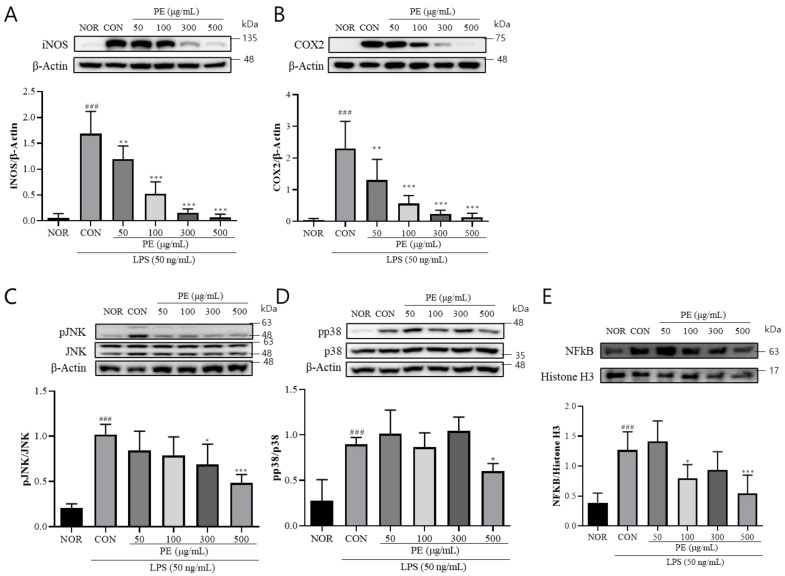
PE suppresses LPS-induced activation of a inflammatory signaling pathway in BV2 microglial cells. BV2 microglial cells were pretreated with PE at indicated concentrations (50, 100, 300, and 500 μg/mL) for 2 h, followed by stimulation with LPS (50 ng/mL). Protein expression was analyzed by Western blot. (**A**) iNOS expression normalized to β-actin. (**B**) COX-2 expression normalized to β-actin. (**C**) p-JNK normalized to total JNK. (**D**) pp38 normalized to total p38. (**E**) Nuclear NF-κB expression normalized to histone H3. Data are presented as means ± SDs. Statistical analysis was performed using one-way ANOVA followed by the Newman–Keuls post hoc test. ### *p* < 0.001 vs. normal (NOR) group; * *p* < 0.05, ** *p* < 0.01, and *** *p* < 0.001 vs. LPS-treated control (CON) group.

**Figure 3 pharmaceutics-17-01045-f003:**
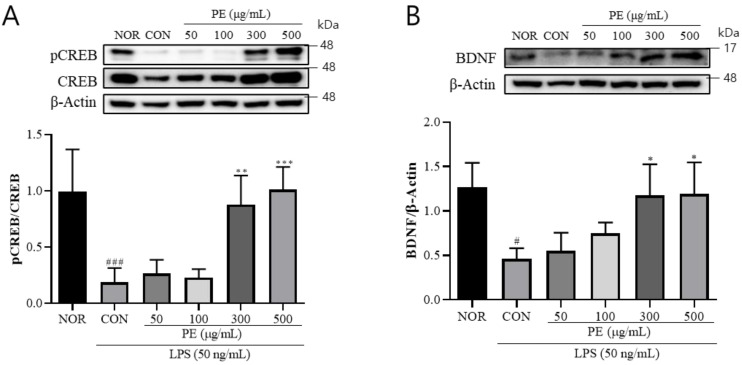
The effect of PE on LPS-induced changes in CREB phosphorylation and BDNF expression in BV2 microglial cells. BV2 microglial cells were pretreated with PE for 2 h, followed by stimulation with LPS (50 ng/mL). Protein expression was analyzed by Western blot. (**A**) pCREB expression normalized to total CREB. (**B**) mature BDNF expression normalized to β-actin. Data are presented as means ± SDs. Statistical analysis was performed using one-way ANOVA followed by the Newman–Keuls post hoc test. # *p* < 0.05, ### *p* < 0.001 vs. normal (NOR) group; * *p* < 0.05, ** *p* < 0.01, and *** *p* < 0.001 vs. LPS-treated control (CON) group.

**Figure 4 pharmaceutics-17-01045-f004:**
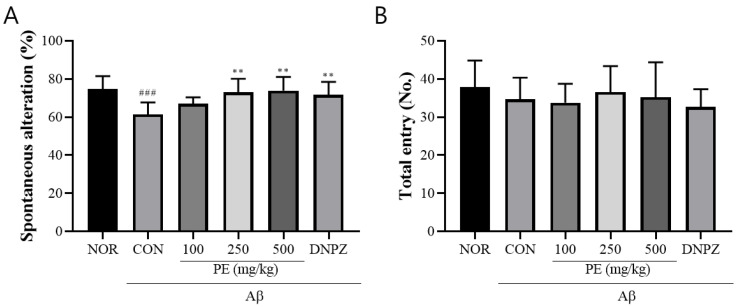
Effect of PE on Aβ-induced memory deficits in the Y-maze test. The Y-maze test was performed for 8 min to evaluate short-term memory. (**A**) Spontaneous alternation, indicating cognitive performance. (**B**) The number of arm entries, reflecting locomotor activity. Data are presented as means ± SDs (n = 9 per group). Statistical analysis was performed using one-way ANOVA followed by the Newman–Keuls post hoc test. ### *p* < 0.001 vs. normal (NOR) group; ** *p* < 0.01 vs. Aβ-injected control (CON) group.

**Figure 5 pharmaceutics-17-01045-f005:**
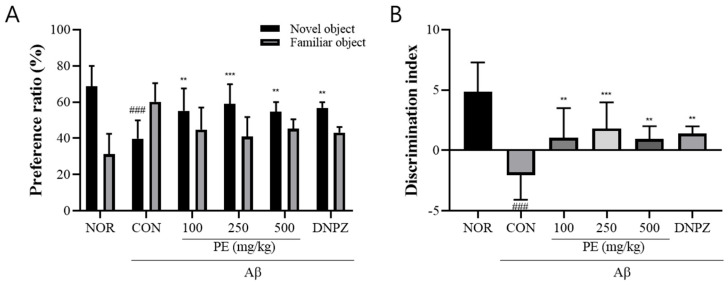
Effect of PE on Aβ-induced memory deficits in the novel object recognition test. The novel object recognition test was conducted to assess recognition memory, and exploration behavior was recorded for 5 min. (**A**) Preference ratio, calculated as the time spent exploring the novel object divided by the total exploration time. (**B**) Discrimination index, reflecting the ability to discriminate between familiar and novel injects. Data are presented as means ± SDs (n = 9 per group). Statistical analysis was performed using one-way ANOVA followed by the Newman–Keuls post hoc test. ### *p* < 0.001 vs. normal (NOR) group; ** *p* < 0.01 and *** *p* < 0.001 vs. Aβ-injected control (CON) group.

**Figure 6 pharmaceutics-17-01045-f006:**
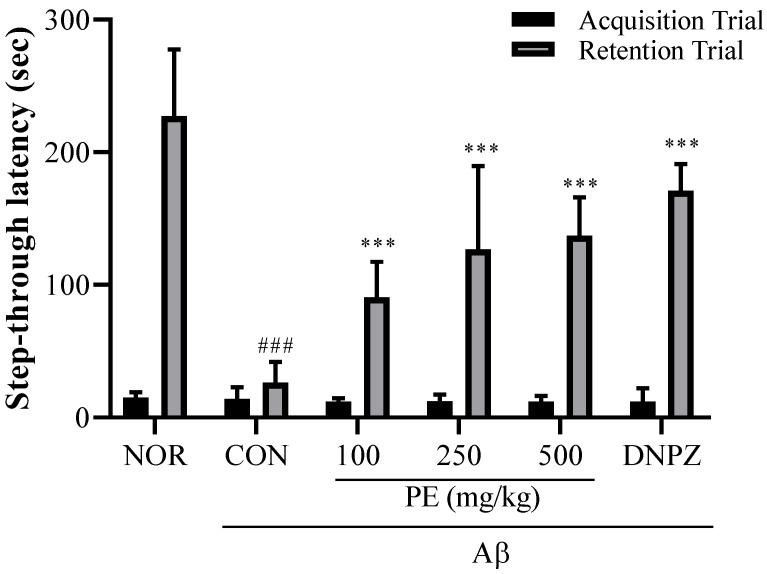
Effect of PE on Aβ-induced memory deficits in the passive avoidance test. The passive avoidance test was used to evaluate associative learning and memory based on aversive stimuli. The step-through latency to enter the dark compartment was measured during the retention trial conducted 24 h after the training session, with increased latency indicating improved memory retention. Data are presented as means ± SDs (n = 10 per group). Statistical analysis was performed using one-way ANOVA followed by the Newman–Keuls post hoc test. ### *p* < 0.001 vs. normal (NOR) group; *** *p* < 0.001 vs. Aβ-injected control (CON) group.

**Figure 7 pharmaceutics-17-01045-f007:**
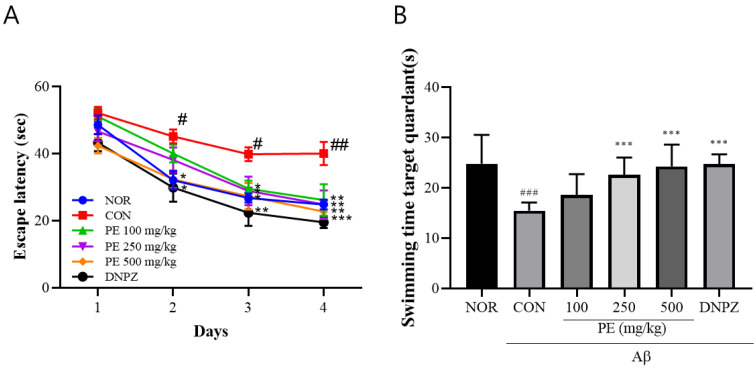
Effects of PE on Aß-induced memory deficits in the Morris water maze test. The Morris water maze test was performed to assess spatial learning and memory abilities. (**A**) Escape latency during the training trials, representing the time required to locate the hidden platform. (**B**) Swimming time in the target quadrant during the probe trial, indicating memory retention of the platform location. Data are presented as means ± SDs (n = 9 per group). Statistical analysis was performed using one-way ANOVA followed by the Newman–Keuls post hoc test. # *p* < 0.05, ## *p* < 0.01, ### *p* < 0.001 vs. normal (NOR) group; * *p* < 0.05, ** *p* < 0.01, and *** *p* < 0.001 vs. Aβ-injected control (CON) group.

**Figure 8 pharmaceutics-17-01045-f008:**
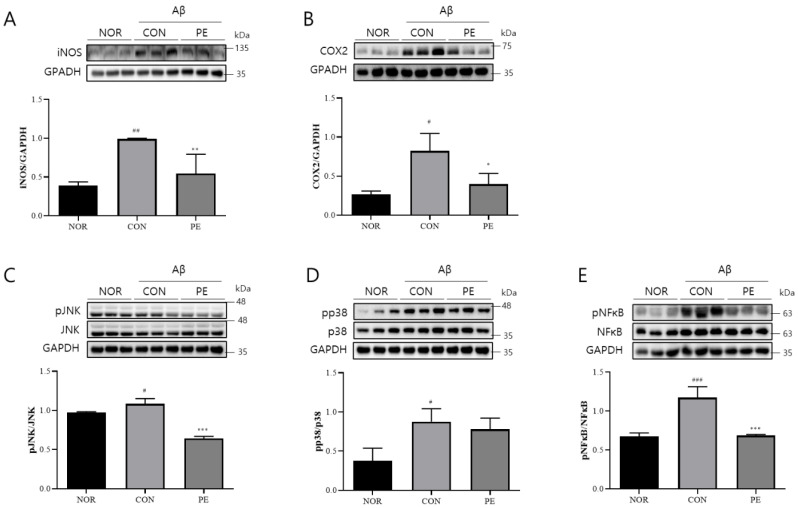
Effects of PE on Aβ-induced neuroinflammation. Western blot analysis was performed on hippocampal tissue from Aβ-injected mice to assess the effect of PE on neuroinflammatory markers. Mice were administered PE (500 mg/kg, p.o.) once daily for 7 d after Aβ injection. (**A**) iNOS expression normalized to GAPDH. (**B**) COX-2 expression normalized to GAPDH. (**C**) pJNK normalized to total JNK. (**D**) pp38 normalized to total p38. (**E**) pNF-κB expression normalized to total NF-κB. Data are presented as means ± SDs. Statistical analysis was performed using one-way ANOVA followed by the Newman–Keuls post hoc test. # *p* < 0.05, ## *p* < 0.01, and ### *p* < 0.001 vs. normal (NOR) group; * *p* < 0.05, ** *p* < 0.01, and *** *p* < 0.001 vs. Aβ-injected control (CON) group.

**Figure 9 pharmaceutics-17-01045-f009:**
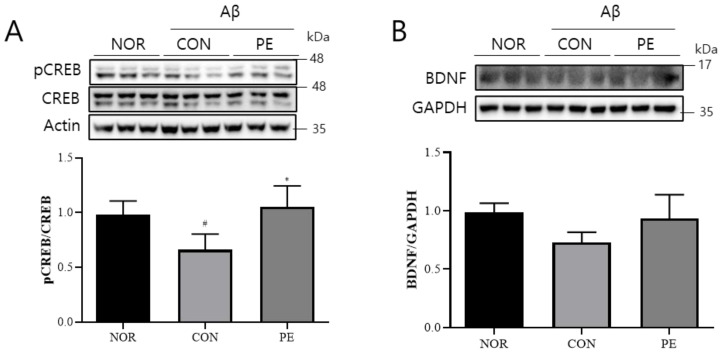
Effect of PE on Aβ-induced changes in pCREB/BDNF signaling. Western blot analysis was performed on hippocampal tissue from Aβ-injected mice to assess the effect of PE on CREB/BDNF signaling. Mice were administered PE (500 mg/kg, p.o.) once daily for 7 d after Aβ injection. (**A**) pCREB normalized to total CREB. (**B**) BDNF expression normalized to GAPDH. Data are presented as means ± SDs. Statistical analysis was performed using one-way ANOVA followed by the Newman–Keuls post hoc test. # *p* < 0.05 vs. normal (NOR) group; * *p* < 0.05 vs. Aβ-injected control (CON) group.

## Data Availability

Data will be made available upon request due to restrictions, e.g., privacy or ethical concerns.

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
