# Peer review of "Anti-Neuroinflammation Effect of Standardized Ethanol Extract of Leaves of *Perilla frutescens* var. *acuta* on Aβ-Induced Alzheimer’s Disease-like Mouse Model"

_pharmaceutics, 2025, doi:10.3390/pharmaceutics17081045_

Round 1
Reviewer 1 Report
Comments and Suggestions for Authors
This is an interesting preclinical study investigating the neuroprotective and anti-inflammatory properties of a standardized ethanol extract of Perilla frutescens leaves (PE) in LPS-induced BV2 microglial cell models and Aβ-induced Alzheimer’s disease (AD)-like mouse models.
Comments:
- The study lacks histological, immunohistochemical, and histomorphometric data to support behavioral and molecular results. For example, the Aβ plaque load and hippocampal morphology were not assessed.
- Since several of the authors are affiliated with the company that financially supported the study, it is important to indicate if blinding procedures were applied in behavioral tests to avoid bias.
- Line 389, correct the sentence '... reduced iNOS and iNOS ...'.
- Lines 503-504 are redundant and can be removed.
Author Response
Comments 1. The study lacks histological, immunohistochemical, and histomorphometric data to support behavioral and molecular results. For example, the Aβ plaque load and hippocampal morphology were not assessed.
Response 1. Thank you very much for your valuable comments. We fully agree that histological, immunohistochemical, and histomorphometric analyses, such as the assessment of Aβ plaque load and hippocampal morphology, would significantly strengthen our findings. However, we regret to inform you that we do not currently have perfusion-fixed brain tissues available for such staining procedures. Given the time frame of 10 days, it is realistically difficult for us to newly conduct these experiments and obtain meaningful results. We kindly ask for your understanding of this limitation, and we hope that the behavioral and molecular data presented in the manuscript are still sufficient to support the conclusions drawn from our study.
Comments 2. Since several of the authors are affiliated with the company that financially supported the study, it is important to indicate if blinding procedures were applied in behavioral tests to avoid bias.
Response 2. Thank you for pointing this out. Blinding procedures were indeed applied during the behavioral tests to minimize potential bias; however, we acknowledge that this important information was inadvertently omitted from the Methods section. We have now revised the manuscript to include a clear statement indicating that the investigators conducting the behavioral assessments were blinded to the treatment conditions (page 5, lines 189-192). We appreciate your careful review and helpful suggestion.
Comments 3. Line 389, correct the sentence '... reduced iNOS and iNOS ...'.
Response 3. Thank you for pointing this out. According to reviewer’s indication, we corrected the sentence in the revised manuscript.
Comments 4. Lines 503-504 are redundant and can be removed.
Response 3. Thank you for pointing this out. According to reviewer’s suggestion, we deleted the sentence in the revised manuscript.
Reviewer 2 Report
Comments and Suggestions for Authors
This manuscript presents a well-executed and comprehensive investigation into the neuroprotective and anti-inflammatory properties of a standardized 60% ethanol extract of Perilla frutescens var. acuta (PE) in both in vitro and in vivo models of Alzheimer’s disease (AD). The authors systematically demonstrate that PE reduces LPS-induced inflammation in BV2 microglial cells and mitigates Aβ-induced cognitive deficits in ICR mice through modulation of key inflammatory and neuroplasticity-related pathways, including MAPKs, NF-κB, CREB, and BDNF. The study is timely and contributes meaningfully to the growing interest in plant-based neuroprotectants for early-stage intervention in AD and mild cognitive impairment (MCI). However, there are several areas where the manuscript could be improved in clarity, scientific rigor, and presentation.
- The rationale behind the selected doses (100, 250, 500 mg/kg) of PE is not adequately explained. Also, were there any observations of toxicity or behavioral side effects in treatment groups?
- The PE extract was standardized to contain rosmarinic acid, yet other bioactives (e.g., scutellarin, luteolin) may contribute to the observed pharmacological effects. Like rosmarinic acid, luteolin is well documented for its neuroprotective, antioxidant, and anti-inflammatory activities. Therefore, it is highly likely that the overall efficacy of PE results from the synergistic or additive actions of multiple phytoconstituents. To substantiate this, the authors are strongly encouraged to perform a detailed phytochemical characterization of the extract using advanced analytical techniques such as LC-MS/MS. Additionally, discussion of such synergy should be expanded in the manuscript to enhance the mechanistic plausibility of PE's neuroprotective action.
- The authors should more clearly articulate how their standardized PE preparation differs from prior studies?
- The relatively short treatment period (7 days) may not reflect the time scale required for therapeutic effects in a real clinical scenario.
- Stereotaxic coordinates, anaesthesia, and postoperative care are missing. Intracerebroventricular injection requires precise co ordinates and anaesthetic regimen
- What was the exact number of animals per group? What was the mortality rate during this study?
- Concentration of Aβ1-42 is missing
- Confirm that sample size justification or power analysis was performed prior to the study.
- Was the behavioral improvement dose-dependent? This can only be substantiated if statistically significant differences are observed between all treatment groups.
Author Response
Comments 1. The rationale behind the selected doses (100, 250, 500 mg/kg) of PE is not adequately explained. Also, were there any observations of toxicity or behavioral side effects in treatment groups?
Response 1. Thank you for your valuable comment. The selection of PE doses (100, 250, and 500 mg/kg) in our study was based on a previous report titled "Effects of Perilla frutescens var. acuta in amyloid β toxicity and Alzheimer's disease-like pathology in 5XFAD mice" (Cho et al., 2017), where doses up to 500 mg/kg were administered without inducing toxicity. We referenced this in the revised manuscript (page 4, lines 184-185). In our study as well, we did not observe any signs of toxicity or behavioral side effects in the treatment groups during the experimental period.
Comments 2. The PE extract was standardized to contain rosmarinic acid, yet other bioactives (e.g., scutellarin, luteolin) may contribute to the observed pharmacological effects. Like rosmarinic acid, luteolin is well documented for its neuroprotective, antioxidant, and anti-inflammatory activities. Therefore, it is highly likely that the overall efficacy of PE results from the synergistic or additive actions of multiple phytoconstituents. To substantiate this, the authors are strongly encouraged to perform a detailed phytochemical characterization of the extract using advanced analytical techniques such as LC-MS/MS. Additionally, discussion of such synergy should be expanded in the manuscript to enhance the mechanistic plausibility of PE's neuroprotective action.
Response 2. We greatly appreciate the reviewer’s valuable suggestion to strengthen the phytochemical characterization of the PE extract and provide a more comprehensive mechanistic explanation of its neuroprotective effects. Although an LC-MS/MS analysis of the final PE extract was not conducted, we performed an in-depth LC-MS/MS-based quantitative profiling of phenolic compounds in the dried leaves of Perilla frutescens var. acuta, the starting material for our standardized extract. This analysis identified and quantified a total of 22 different phenolic compounds, including major bioactives such as rosmarinic acid, luteolin-glucuronide, luteolin-hexoside, quercetin-3-O-glucoside, caffeic acid, and chrysin. Among these, rosmarinic acid was the most abundant (approximately 57–67 mg/g dry leaf), with substantial levels of luteolin derivatives and other flavonoids also present. These findings strongly suggest that the PE extract contains not only rosmarinic acid but also a diverse range of polyphenols and flavonoids, notably luteolin and its glycosides, which are well documented for their neuroprotective, anti-inflammatory, and antioxidant effects. Thus, the cognitive and anti-neuroinflammatory benefits observed in our study are likely attributable to the synergistic or additive actions of these multiple phytoconstituents. Accordingly, we have revised the Discussion section to provide a clearer account of the phytochemical complexity of Perilla leaves and to elaborate on potential synergistic mechanisms involving rosmarinic acid, luteolin, and other co-existing phenolics. While further studies are warranted to directly profile the final PE extract and delineate the contributions of individual components, our current phytochemical data increase the plausibility that the therapeutic efficacy of PE arises from the combined actions of these bioactive compounds (page 14, lines 524-530).
Comments 3. The authors should more clearly articulate how their standardized PE preparation differs from prior studies?
Response 3. Thank you for your comment. In contrast to previous studies, the PE used in our study was extracted using 60% ethanol at 50°C, a method selected to enable potential clinical application and large-scale production. This standardized extraction condition was optimized to ensure both reproducibility and scalability, making it more suitable for future translational research.
Comments 4. The relatively short treatment period (7 days) may not reflect the time scale required for therapeutic effects in a real clinical scenario.
Response 4. We fully agree with your insightful comment. In this study, due to the constraints of evaluating the efficacy of multiple candidate compounds within a limited timeframe, we opted for a relatively short treatment duration of 7 days. Moreover, the animal model used here is an acute model in which Aβ₁₋₄₂aggregates are directly injected into the cerebral ventricle, allowing us to observe drug effects within a short period. For long-term effects, our previous studies using the 5XFAD transgenic mouse model have already demonstrated the efficacy of extended treatment. In the current study, our primary objective was to evaluate the anti-inflammatory potential of PE extracted through a novel method, and we found that even short-term administration was sufficient to observe meaningful effects. Nonetheless, we fully agree with your point that studies employing longer treatment durations, which better reflect clinical scenarios, are necessary to draw more comprehensive conclusions. We will take this into account in future investigations.
Comments 5. Stereotaxic coordinates, anaesthesia, and postoperative care are missing. Intracerebroventricular injection requires precise co-ordinates and anaesthetic regimen
Response 5. Thank you for your valuable comment. We appreciate your attention to the experimental details. In response, we have clarified the anesthesia and injection procedures in the revised manuscript. Specifically, we now state that mice were anesthetized with isoflurane (3% for induction and 1.5% for maintenance), and Aβ₁₋₄₂ aggregates were injected into the third ventricle using stereotaxic coordinates (AP: −2.00 mm, ML: 0 mm, DV: −2.00 mm). This information has been added to the Materials and Methods section of the revised manuscript (page 4, lines 180-183).
Comments 6. What was the exact number of animals per group? What was the mortality rate during this study?
Response 6. Thank you for your question. The passive avoidance test was conducted with 10 animals per group, while the other behavioral tests were performed with 9 animals per group. No mortality was observed in any group during the course of the study. In certain experiments, animals that did not meet behavioral performance criteria were excluded from the analysis, resulting in n=9 for some groups. Specifically, animals were excluded if they exhibited minimal activity during the tests, defined as follows: total arm entries ≤5 in the Y-maze test, object exploration time ≤5 seconds in the object recognition test, step-through latency ≥60 seconds in the passive avoidance test, and floating without active swimming in the Morris water maze. We included this in the revised manuscript (page 6, lines 235-238).
Comments 7. Concentration of Aβ1-42 is missing
Response 7. Thank you for your comment and for pointing out the missing information. Aβ1-42 (1 mg/mL, 222 μM) was administered via intracerebroventricular injection at a volume of 5 μL per mouse. We have included this information in the revised manuscript (page 4, line 178).
Comments 8. Confirm that sample size justification or power analysis was performed prior to the study.
Response 8. For the approval of the animal experiment protocol, we conducted a G*Power analysis, which indicated a required sample size of 324 animals. However, based on our extensive research experience over the years, we emphasized that the efficacy of the test compound can be sufficiently confirmed with approximately 9-10 animals per group.
Comments 9. Was the behavioral improvement dose-dependent? This can only be substantiated if statistically significant differences are observed between all treatment groups.
Response 9. Thank you very much for your insightful comment. At the tested doses, a dose-dependent effect was observed in the Y-maze test and Morris water maze test. However, such dose-dependency was not clearly demonstrated in the object recognition and passive avoidance tests. We believe that this trend may become more evident if additional lower-dose experiments are conducted. Unfortunately, due to time constraints and the logistical limitations of conducting new animal experiments within the 10-day revision period, it is not feasible to include these additional data at this time. We kindly ask for your understanding in this matter.

Reviewer 3 Report
Comments and Suggestions for Authors
In this study, the authors evaluated the anti-neuroinflammatory potential of a 60% ethanolic extract of Perilla leaves (PE). The authors found that PE significantly suppressed LPS-induced production of nitric oxide and pro-inflammatory mediators, including IL-6, TNF-α, NF-κB, iNOS, and COX-2, along with inhibition of JNK and p38 MAPK activation. Furthermore, PE increased the expression of CREB and BDNF.
The manuscript is well written and well argued, and it is of great interest to the scientific community. The references are appropriate, and the results are clearly presented with thorough statistical analysis. The work deserves to be published in its current form.
Just a few questions out of curiosity:
- Have the authors evaluated the effect of PE on Aβ aggregation?
- Did the authors investigate the effect of metals—particularly copper—on Aβ and the neuroinflammatory process in this study?
Author Response
Comments 1. Have the authors evaluated the effect of PE on Aβ aggregation?
Response 1. We conducted ThT assay for investigating the effect of PE on Aβ aggregation. It potently inhibited Aβ aggregation in a concentration dependent manner.
Figure 1. The effect of PE on Aβ aggregation.
Comments 2. Did the authors investigate the effect of metals—particularly copper—on Aβ and the neuroinflammatory process in this study?
Response 2. Thank you for your insightful question. However, we did not investigate the effects of metals, including copper, on Aβ or the neuroinflammatory process in this study. This is indeed an important and interesting aspect, and we agree that future studies addressing the role of metal ions in Alzheimer's disease pathology would be valuable.

Round 2
Reviewer 2 Report
Comments and Suggestions for Authors
The authors have addressed all the queries raised. The manuscript may be accepted.